

# minSNPs: an R package for the derivation of resolution-optimised SNP sets from microbial genomic data

Kian Soon Hoon[1], Deborah C. Holt[1,2], Sarah Auburn[1,3,4], Peter Shaw[5] and Philip M. Giffard[1,2]

[1] Menzies School of Health Research, Charles Darwin University, Darwin, Northern Territory, Australia
[2] CDU Menzies School of Medicine, Faculty of Health, Charles Darwin University, Darwin, Northern Territory, Australia
[3] Mahidol-Oxford Tropical Medicine Research Unit, Mahidol University, Bangkok, Thailand
[4] Centre for Tropical Medicine and Global Health, University of Oxford, Oxford, United Kingdom
[5] Oujian Laboratory, Wenzhou, Zhejiang, China

Corresponding author
Philip M. Giffard,
phil.giffard@menzies.edu.au

## ABSTRACT

Here, we present the R package, minSNPs. This is a re-development of a previously described Java application named Minimum SNPs. MinSNPs assembles resolution-optimised sets of single nucleotide polymorphisms (SNPs) from sequence alignments such as genome-wide orthologous SNP matrices. MinSNPs can derive sets of SNPs optimised for discriminating any user-defined combination of sequences from all others. Alternatively, SNP sets may be optimised to determine all sequences from all other sequences, *i.e.*, to maximise diversity. MinSNPs encompasses functions that facilitate rapid and flexible SNP mining, and clear and comprehensive presentation of the results. The minSNPs' running time scales in a linear fashion with input data volume and the numbers of SNPs and SNPs sets specified in the output. MinSNPs was tested using a previously reported orthologous SNP matrix of *Staphylococcus aureus* and an orthologous SNP matrix of 3,279 genomes with 164,335 SNPs assembled from four *S. aureus* short read genomic data sets. MinSNPs was shown to be effective for deriving discriminatory SNP sets for potential surveillance targets and in identifying SNP sets optimised to discriminate isolates from different clonal complexes. MinSNPs was also tested with a large *Plasmodium vivax* orthologous SNP matrix. A set of five SNPs was derived that reliably indicated the country of origin within three south-east Asian countries. In summary, we report the capacity to assemble comprehensive SNP matrices that effectively capture microbial genomic diversity, and to rapidly and flexibly mine these entities for optimised marker sets.

## INTRODUCTION

The extremely large-scale accumulation of microbial whole genome sequence information provides a potent resource for the design of targeted genetic analysis procedures. Whole genome analysis is now widely applied directly to public health, clinical, and research microbiology. However, targeted genetic analyses may be complementary

to whole genome analysis for purposes such as high-volume, low-cost surveillance, primary clinical or environmental specimen analysis, and analyses performed outside the laboratory environment. Several research groups have recently developed single nucleotide polymorphisms (SNP)-based genotyping approaches, *e.g.*, to investigate *Mycobacterium* species (*Kim et al., 2021*; *Napier et al., 2020*), attribute hosts for *Chlamydia psittaci* (*Vorimore et al., 2021*) and *Campylobacter coli* (*Jehanne et al., 2020*), distinguish *Rickettsia typhi* from different continents (*Kato et al., 2022*), identify *Escherichia coli* of specific serotype (*Rahman, Lim & Park, 2022*), and track the spread of drug resistance in *Plasmodium falciparum* infections (*Jacob et al., 2021*).

Reported approaches to the selection of the SNP sets used in such methods are varied and reflect the purpose of genotyping. SNP sets with high generalised discriminatory power may be identified on the basis of high minor allele frequency (*Fola et al., 2020*). SNPs for geographic region source attribution can be identified as the basis of the fixation index ($F_{ST}$), which in essence, is the difference in allele frequencies between isolates from different regions. This can be combined with machine learning (*Trimarsanto et al., 2022*). Strain-specific SNPs can be identified using several approaches, with an example being the wgSNP module in the commercial Bionumerics software (*Vorimore et al., 2021*). SNP sets identified with all these approaches may potentially be combined with known functional SNPs that confer phenotypes of interest, such as non-susceptibility to antimicrobial agents.

Here we report the R package "minSNPs". This is designed to derive sets of SNPs from biological sequence alignment data on the basis of high combinatorial discriminatory power. The envisioned application is the derivation of high-resolution sets of SNPs from DNA sequence alignments or orthologous SNP matrices. minSNPs encompasses much of the functionality of the previously reported "Minimum SNPs" Java-based bioinformatics application (*Robertson et al., 2004*; *Price et al., 2007*). Minimum SNPs was used to develop several SNP-based bacterial genotyping methods, *e.g.*, (*Tong et al., 2011*; *Price et al., 2007*; *Giffard et al., 2018*; *Holt et al., 2021*; *Lilliebridge et al., 2011*). minSNPs is a new package, written in R, with distinct code from Minimum SNPs. The reasons for re-development were improvement of flexibility, error handling, and output formats. More specifically, with Minimum SNPs, identifying SNPs diagnostic for groups of sequence variants is laborious, there is no facility to check and amend input files to ensure they are analysable, and the output formats do not provide complete tabulated information regarding the relationship between SNP allele profiles and the input sequences. Further, minSNPs is an R package (as opposed to Java). It is now available in the usual public repositories, consistent with current trends and conventions for academic software in the data sciences. Also, unlike Java software, it is straightforward to make R packages available on UNIX-based computer clusters. To our knowledge, complete minSNPs functionality is not found in any other software for SNP set derivation.

Here we describe minSNPs and demonstrate functionality using comparative genome data from *Staphylococcus aureus* and *P. vivax*. We also demonstrate using minSNPs with input files generated from multiple short read data sets.

## METHOD & IMPLEMENTATION

The input format for minSNPs is a sequence alignment in FASTA format. All symbols can be recognised so that the program can derive sets of polymorphic positions from any file in a FASTA format alignment, irrespective of the symbols in the sequences. However, the default state is that symbols that are not G, A, T, or C trigger the exclusion of the relevant alignment positions from the analysis. The characters that do not trigger exclusion can be defined by the user. While minSNPs does not support input of other file formats, there are tools to extract and convert VCF or other common formats to FASTA, including vcf2phylip.py (*Ortiz, 2019*) and vcftools (*Danecek et al., 2011*). We demonstrate the use of vcf2phylip.py in the "Derivation of *Plasmodium vivax* SNP sets" section.

The output of minSNPs is set(s) of polymorphic positions in the alignment. SNP sets are assembled iteratively on the basis of maximised combinatorial resolving power. In other words, the program scans all acceptable positions to identify the SNP that confers the maximum discriminatory power in combination with SNPs already in the SNP set (if any). This SNP is added to the set. Where more than one SNP confers the same increase in resolving power, the SNP nearest to position 1 of the alignment will be added to the set.

There are two user-selectable algorithms for measuring resolving power.

1. **Percent mode.** The resolving power is the percentage of sequences in the alignment that are not discriminated from the user-selected sequence(s) (the group of interest). The SNP sets are constrained to 100% sensitivity. The first SNP identified is the 100% sensitive SNP with maximum possible specificity. Subsequent SNPs are selected on the basis of the maximum possible increase in specificity in combination with the previously selected SNP(s). All alignment positions that are variable within the group of interest can optionally be excluded from the analysis. This has no effect on the search algorithm for two-state SNPs but can affect searches involving three-state SNPs. We suggest that, where possible, the group of interest be composed of > 1 sequence to avoid the identification of spurious SNPs arising from sequencing errors.

2. **Simpson mode**. The resolving power is the power to discriminate "all from all", as measured by the Simpsons index of diversity. In this context, the index of the diversity is the probability that any two sequences in the alignment will be discriminated from each other by the SNP set, as calculated by index of the diversity $= 1 - \frac{1}{N(N-1)} \sum_{j=1}^{s} n_j(n_j - 1)$, where N is the number of sequences, s is the number of classes defined by the SNPs, and $n_j$ is the number of sequences defined by the class j (*Robertson et al., 2004*).

In the main search function, the user specifies the size and number of the SNP sets that constitute the output. When multiple SNP sets are requested, minSNPs identifies alternative SNP sets that are all resolution optimised, with the constraint that the sets must differ from each other, at least in the first SNP. The user can force the program to include or exclude any alignment position(s) in/from the SNP set. Where positions are included, new SNPs are identified based on resolving power in combination with the included positions. This facilitates rapid exploration of SNP sets.

minSNPs can identify alignment positions where at least one sequence has a non-standard DNA symbol, and these positions are optionally excluded from the analysis. Indels

(dashes) default to being regarded as symbols equivalent to other symbols. Alternatively, the user can specify that indels trigger the exclusion of the relevant alignment positions from the analysis. There is also an optional function to exclude positions with SNPs with > 2 alleles.

minSNPs provides a cumulative increase in resolving power as the sets are built, and the tabulated information indexing the sequences in the alignment as defined by each allelic profile. For percent mode analyses, this is within a "group of interest or non-group of interest" framework. The outputs are presented in the R console and optionally outputted to a tab-delimited format file. A facile method to fully define the informative power of a SNP set derived by percent analysis is to force the inclusion of the identified SNPs into a Simpson mode analysis, in which the user-defined SNP set size equals the number of included SNPs, *i.e.,* no additional SNPs are derived. This will reveal how the sequences assort in relation to allelic profiles of the "forced included" SNPs. Alternatively, this can be done in reverse to assess the performance of a Simpson's index of diversity maximised SNP set to detect user-defined subsets of sequences with 100% sensitivity. These strategies provide considerable flexibility regarding the exploration of SNP sets.

Package access and documentation are described in the "Availability" section. An abbreviated user guide and the search algorithm are in Fig. 1.

## RESULTS AND DISCUSSION

To explore the potential utility of minSNPs, we:
1. Determined the relationship between input alignment dimensions and the number and size of output SNP sets, with running time;
2. Generated SNP sets of potential relevance to surveillance from orthologous SNP matrices derived from genomic epidemiology studies in *S. aureus* and *P. vivax* and;
3. Generated single orthologous SNP matrices from multiple short-read data sets to demonstrate the utility of minSNPs for analysing large-scale comparative genome data from multiple studies.

### Run-time determinations
The relationships between the analysis time and dimensions of the input alignment, the number of SNPs in the output SNP set, and the number of SNP sets in the output were determined. The relationship was linear with respect to all three parameters. Examples of running time are shown in Table 1. It was also demonstrated that running minSNPs using multiple cores improves its performance. Complete data and code are shown at figshare (https://doi.org/10.6084/m9.figshare.19579816.v1). The faster run-time on a laptop when compared to a high-performance cluster (HPC) was due to the simpler architecture of the machine; we note that when the dimension of the alignments increases, the HPC's comparative performance improves. Therefore, given a higher number of cores and increased memory available, an HPC can outperform a laptop. Our general experience is that minSNPs can be readily used for substantial analyses on PCs with an Intel i5-7500T CPU running at 2.70 GHz,and with 8 Gb of RAM, and 236 Gb of storage.
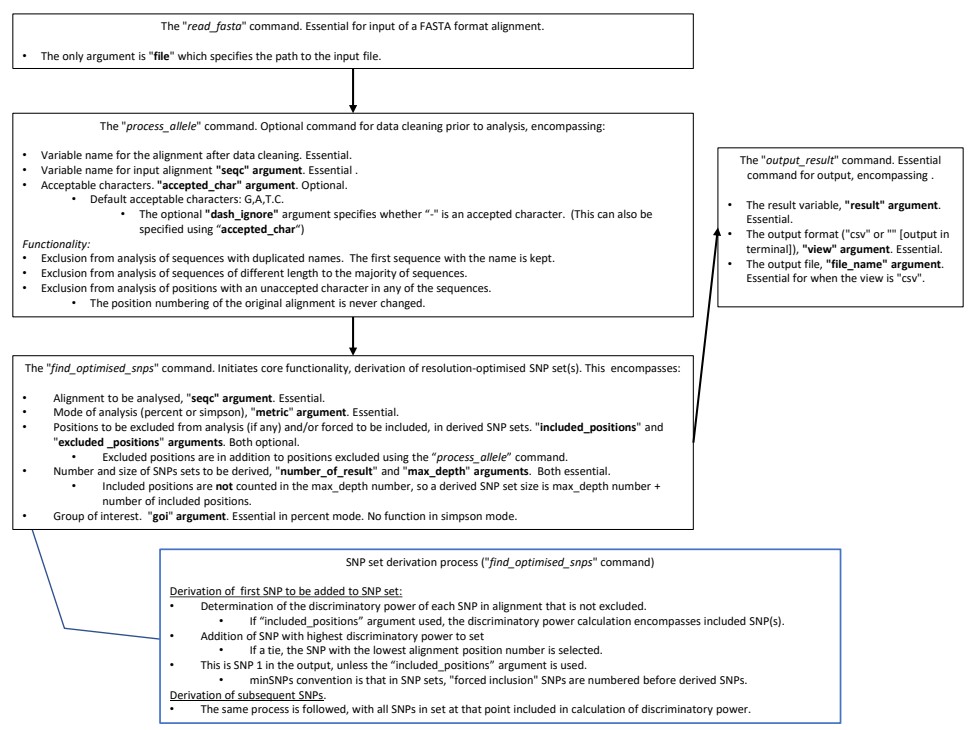

**Figure 1** A summary of how to use minSNPs, and the SNP search algorithm.

**Table 1** **Input alignment dimensions versus run time.** "Percent" mode stops the search once the group of interest is completely discriminated, hence increasing the number of SNPs in SNP set does not necessarily increase the running time. The laptop used to test the package consisted of an AMD Ryzen 7 4800H processor and 16GB RAM; we have found that other lower specs laptop will have no problem running minSNPs analysis for alignment of this size.

| Mode | Input alignment dimensions | Number of SNPs in SNP set | Running time HPC (s) | | Running time Laptop (s) | |
|---|---|---|---|---|---|---|
| | | | 2 Cores | 8 cores | 2 cores | 8 cores |
| Percents | 167 isolates; 1000 SNPs | 1 | 1.798s | 1.889s | 0.816s | 0.703s |
| | 167 isolates; 10,000 SNPs | 1 | 15.862s | 10.166s | 6.649s | 2.929s |
| | 50 isolates; 20,651 SNPs | 1 | 9.673s | 7.480s | 4.438s | 2.024s |
| | 167 isolates; 20,651 SNPs | 1 | 32.874s | 17.339s | 16.147s | 5.977s |
| Simpson | 167 isolates; 1000 SNPs | 1 | 1.761s | 1.571s | 0.863s | 0.595s |
| | 167 isolates; 10,000 SNPs | 1 | 15.193s | 10.913s | 8.145s | 3.213s |
| | 50 isolates; 20,651 SNPs | 1 | 10.144s | 7.452s | 5.972s | 2.350s |
| | 167 isolates; 20,651 SNPs | 1 | 32.697s | 19.203s | 16.687s | 6.475s |
| | | 3 | 93.186s | 60.749s | 49.029s | 21.662s |
| | | 5 | 157.831s | 105.136s | 85.098s | 35.363s |

## Derivation of SNP sets from a *Staphylococcus aureus* orthologous SNP matrix

To demonstrate minSNPs' functionality, we analysed genome-wide orthologous SNP matrices to identify SNP sets diagnostic for a conserved lineage that is a potential
surveillance target, SNP sets diagnostic for a broader phylogenetic lineage that encompasses the potential surveillance target, and SNP sets optimised with respect to Simpson's index of diversity. For the latter, our interests were in the resolving power (the Simpson's index of diversity) and the concordance of the genotypes defined by the SNP sets with the phylogeny indicated by the orthologous SNP matrix.

We first analysed a previously described orthologous SNP matrix (*Holt et al., 2021*, S4 Data. Orthologous SNP matrix) composed of 20,651 SNPs from 162 *S. aureus* isolates, four *Staphylococcus argenteus* isolates, and *S. aureus* Mu50, which was the reference genome for matrix construction (*Holt et al., 2021*). The isolates were from a study in the north of the Australian Northern Territory, revealing potential *S. aureus* transmission events involving haemodialysis patients and potential contacts in the clinical context (STARRS study) (*Holt et al., 2021*).

### Derivation of SNP sets to discriminate ST762 with "percent" mode

The STARRS study identified isolates of multilocus sequence typing (MLST) defined ST762 (clonal complex (CC) 1) and were involved in transmission events leading to patient infections. ST762 is vanishingly rare globally but was prevalent in the STARRS study. We, therefore, used the ST762 lineage as a model for a potential surveillance target. Using minSNPs in percent mode, we determined that 12 SNPs each individually discriminated all the ST762 isolates from other isolates in the study, with 100% sensitivity and specificity (figshare: https://doi.org/10.6084/m9.figshare.19579837.v1). A BLAST analysis demonstrated that for each of these SNPs, the alleles present in the ST762 isolates were not present in the public databases, suggesting that these SNPs have a generalised ability to discriminate ST762 from the remainder of the *S. aureus* complex (figshare: https://doi.org/10.6084/m9.figshare.19579831.v1).

### Derivation of SNP sets to discriminate CC1 with "percent" mode

The same procedure was used to derive SNP sets that discriminate the CC1 (ST1 and ST762) STARRS isolates from the other isolates. It was found that there were 119 SNPs that each individually provided 100% sensitivity and specificity for CC1 isolates (figshare: https://doi.org/10.6084/m9.figshare.19579837.v1). Similar to SNPs identified for ST762, a BLAST analysis returned 61 specimens from Genbank; out of these, 53 were CC1, with three false positives belonging to ST425 and five specimens untypeable by MLST (figshare: https://doi.org/10.6084/m9.figshare.19579831.v1).

### Derivation of SNP sets with "Simpson" mode

We further used minSNPs to derive 15 five-member SNP sets with maximised Simpson's index of diversity. The index of diversity values obtained ranged from 0.925 to 0.936, defining 16 to 21 genotypes. Concordance with phylogeny was determined for two SNP sets (set 1 and 11) that were selected based on having no SNPs in common. Both SNP sets discriminated the major lineages defined by the STARRS SNP matrix (Fig. 2, Table 2).

### Derivation of *Plasmodium vivax* SNP sets

Given the challenges associated with the large genome size and high proportions of 'contaminating' human DNA, targeted SNP genotyping remains an important approach

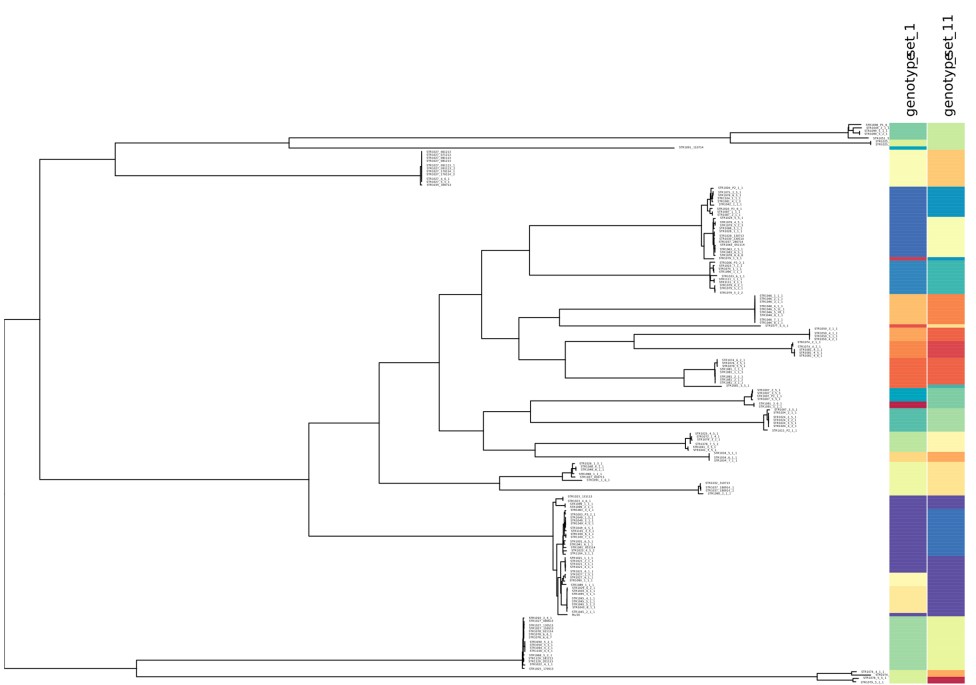

**Figure 2  Correspondence between SNP allele genotypes and phylogeny for the *S. aureus* STARRS data.**
Correspondence between SNP allele genotypes and phylogeny for the *S. aureus* STARRS data. The phylogenetic tree was reproduced from (15) and labelled with two newly identified high-D SNP sets (https://microreact.org/project/minsnps-starrs). High-diversity index SNP sets 1 and 11 are comprised of positions 111760, 1925985, 2663300, 2683490, 124088, and position 539419, 1413096, 1146945, 2184528, 1577370, of the Mu50 reference genome.

in *Plasmodium* epidemiological tracking (*Fola et al., 2020*; *Noviyanti et al., 2020*; *Diez Benavente et al., 2020*). minSNPs was tested with a *P. vivax* orthologous SNP matrix encompassing 259 isolates and 527,107 SNPs (*Auburn et al., 2018*). The matrix is available at the Malariagen website (*Auburn et al., 2018*). This encompasses heterozygote positions, designated by standard nucleotide ambiguity codes, that are the result of polyclonal infections.

The data were generated from isolates collected from Malaysia, Thailand, and Indonesia as part of a study to identify changes in the *P. vivax* population as Sabah (Malaysia) approaches the elimination of vivax malaria (*Auburn et al., 2018*). In 183,509 of the SNPs, a nucleotide ambiguity code (where calls were heterozygote) was assigned to at least one of these isolates. The detailed procedures and all the data for the *P. vivax* experiments are in https://doi.org/10.6084/m9.figshare.19580299.v1. The procedures and results that can be presented concisely are in figshare: https://doi.org/10.6084/m9.figshare.19580299.v1 Overview. Large data sets are in the other files, including this https://figshare.com/s/db47a069aab93f3c615c?file=36141051, which shows the mapping between the SNPs and the reference genome.

### Derivation of SNP sets to discriminate "K2" strain with "percent" mode

A subset of 26 specimens from Malaysia were near identical. These were denoted "K2" strains reflecting isolates that were potentially undergoing clonal expansion (*Auburn et al., 2018*). We regarded these as model surveillance targets. SNPs that discriminated the K2 lineage were identified with minSNPs in percent mode, with all the K2 specimens defined as the group of interest. All 183,509 positions where any of the sequences had an ambiguity code were excluded from the analysis. The results obtained after the analysis of 343,598 SNPs yielded 124 SNPs that each individually discriminated the K2 lineage from all the other isolates in the matrix. These are shown in this figshare: https://doi.org/10.6084/m9.figshare.19580299.v1. Any of these 124 SNPs could potentially form the basis of a K2 surveillance protocol. Using more than one of these SNPs may provide useful redundancy to avoid false negatives due to undiscovered sequence diversity.

### Derivation of SNP sets to discriminate Malaysian strain with "percent" mode with ambiguity codes excluded

Next, SNPs that discriminated all Malaysian specimens from all other specimens were derived. To streamline the analysis, only one of the K2 specimens was included. Initially, we confined the analysis to the 343,598 SNPs that do not encompass any ambiguity codes. This test was not successful. The maximum percent score obtained from five SNPs was 0.265, meaning that 73.5% of the non-Malaysian specimens were not discriminated from the Malaysian specimens. The complete results are in this figshare: (https://doi.org/10.6084/m9.figshare.19580299.v1).

### Derivation of SNP sets to discriminate Malaysian strain with "percent" mode with ambiguity codes transformed

A different protocol was then adopted. Before minSNPs analysis, ambiguity codes were transformed into the major allele of the sequences. In all cases, the major allele was consistent with the ambiguity code. After minSNPs analysis, the relationship between the allelic profiles and isolate was determined using the untransformed matrix. The untransformed matrix can define allelic profiles that include ambiguity codes. Any specimens that had such an allelic profile, *i.e.,* they had an ambiguity code at a SNP within the SNP set being assessed, were classified as untypeable by that SNP set. Typeability was, therefore, a criterion used for assessing SNP sets, although we note that typeability is likely a function of specimen quality and/or whether the specimen contained a mixture of strains. It is not an inherent property of a pure *P. vivax* clone.

Transforming ambiguity codes to the major allele present at each relevant position proved a successful approach to identifying SNPs that discriminated against Malaysian specimens. The complete results are in this figshare: (https://doi.org/10.6084/m9.figshare.19580299.v1). Two sets of two SNPs were identified, each of which discriminated all Malaysian specimens from all other typable specimens. For one SNP set, 20 specimens (7.72%) were untypeable, and for the other, 22 (8.49%). All the Malaysian specimens were typable with both SNP sets. The precise reason for the superior result from the matrix with ambiguity codes transformed has not been determined but is explicable from the

Hoon et al. (2023), *PeerJ*, DOI 10.7717/peerj.15339

**Table 2  STARRS: Breakdown of CC/singletons for genotypes defined by SNP sets 1 and 11.** The distinction between singletons and CCs is somewhat arbitrary. The CCs labelled with an asterisk "*" were present only as the CC founder ST in the STARRS isolates. Column SA refers to *S. argenteus*. Column NA refers to isolates with unknown CC/ST.

**Breakdown of CC/Singletons for genotypes defined by SNPs set 1**

| Genotype | SNPs set 1 (111760, 124088, 1925985, 2663300, 2683490) CC | | | | | | | | | | | | | | | | | |
| | 1 | 5 | 6 | 8 | 12* | 15 | 20* | 30 | 45 | 72 | 78 | 93 | 97 | 101* | 121* | 834 | SA | NA |
|---|---|---|---|---|---|---|---|---|---|---|---|---|---|---|---|---|---|---|
| 1 | 21 | 0 | 0 | 0 | 0 | 0 | 0 | 0 | 0 | 0 | 0 | 0 | 0 | 0 | 0 | 0 | 0 | 0 |
| 2 | 1 | 0 | 0 | 0 | 0 | 0 | 0 | 0 | 0 | 0 | 0 | 0 | 0 | 0 | 0 | 0 | 0 | 0 |
| 3 | 0 | 24 | 0 | 0 | 0 | 0 | 0 | 0 | 0 | 0 | 0 | 0 | 0 | 0 | 0 | 0 | 0 | 0 |
| 4 | 0 | 8 | 0 | 0 | 0 | 0 | 0 | 0 | 0 | 0 | 0 | 0 | 0 | 0 | 0 | 0 | 0 | 0 |
| 5 | 0 | 4 | 0 | 0 | 0 | 0 | 0 | 0 | 0 | 0 | 0 | 0 | 0 | 0 | 0 | 0 | 0 | 0 |
| 6 | 0 | 0 | 5 | 0 | 0 | 0 | 0 | 0 | 0 | 0 | 0 | 0 | 0 | 0 | 0 | 0 | 0 | 0 |
| 7 | 0 | 0 | 0 | 10 | 0 | 0 | 0 | 0 | 0 | 0 | 0 | 0 | 0 | 0 | 0 | 0 | 0 | 0 |
| 8 | 0 | 0 | 0 | 0 | 9 | 0 | 0 | 0 | 0 | 0 | 0 | 0 | 0 | 0 | 0 | 0 | 0 | 0 |
| 9 | 0 | 0 | 0 | 0 | 0 | 10 | 0 | 0 | 0 | 0 | 0 | 0 | 0 | 0 | 0 | 0 | 0 | 0 |
| 10 | 0 | 0 | 0 | 0 | 0 | 0 | 4 | 0 | 0 | 0 | 0 | 0 | 0 | 0 | 0 | 0 | 0 | 0 |
| 11 | 0 | 0 | 0 | 0 | 0 | 0 | 1 | 0 | 0 | 0 | 0 | 0 | 0 | 0 | 0 | 0 | 0 | 0 |
| 12 | 0 | 0 | 0 | 0 | 0 | 0 | 0 | 1 | 0 | 0 | 4 | 0 | 0 | 0 | 0 | 0 | 0 | 0 |
| 13 | 0 | 0 | 0 | 0 | 0 | 0 | 0 | 0 | 5 | 0 | 0 | 0 | 0 | 0 | 0 | 0 | 0 | 0 |
| 14 | 0 | 0 | 0 | 0 | 0 | 0 | 0 | 0 | 2 | 0 | 0 | 0 | 0 | 0 | 0 | 0 | 4 | 0 |
| 15 | 0 | 0 | 0 | 0 | 0 | 0 | 0 | 0 | 0 | 8 | 0 | 0 | 0 | 0 | 0 | 0 | 0 | 1 |
| 16 | 0 | 0 | 0 | 0 | 0 | 0 | 0 | 0 | 0 | 0 | 2 | 0 | 0 | 0 | 0 | 0 | 0 | 0 |
| 17 | 0 | 0 | 0 | 0 | 0 | 0 | 0 | 0 | 0 | 0 | 0 | 16 | 0 | 0 | 0 | 0 | 0 | 0 |
| 18 | 0 | 0 | 0 | 0 | 0 | 0 | 0 | 0 | 0 | 0 | 0 | 0 | 6 | 0 | 0 | 0 | 0 | 0 |
| 19 | 0 | 0 | 0 | 0 | 0 | 0 | 0 | 0 | 0 | 0 | 0 | 0 | 0 | 7 | 0 | 0 | 0 | 0 |
| 20 | 0 | 0 | 0 | 0 | 0 | 0 | 0 | 0 | 0 | 0 | 0 | 0 | 0 | 0 | 11 | 0 | 0 | 0 |
| 21 | 0 | 0 | 0 | 0 | 0 | 0 | 0 | 0 | 0 | 0 | 0 | 0 | 0 | 0 | 0 | 3 | 0 | 0 |

Hoon et al. (2023), *PeerJ*, DOI 10.7717/peerj.15339

**Table 2** (*continued*)

Breakdown of CC/Singletons for genotypes defined by SNPs set 1

| Genotype | SNPs set 1 (111760, 124088, 1925985, 2663300, 2683490) | | | | | | | | | | | | | | | | | |
| | CC | | | | | | | | | | | | | | | | | |
| | 1 | 5 | 6 | 8 | 12* | 15 | 20* | 30 | 45 | 72 | 78 | 93 | 97 | 101* | 121* | 834 | SA | NA |

Breakdown of CC/Singletons for genotypes defined by SNPs set 11

| Simpson | SNPs set 11 (539419, 1146945, 1413096, 1577370, 2184528) | | | | | | | | | | | | | | | | | |
| | CC | | | | | | | | | | | | | | | | | |
| | 1 | 5 | 6 | 8 | 12* | 15 | 20* | 30 | 45 | 72 | 78 | 93 | 97 | 101* | 121* | 834 | SA | NA |
| 1 | 12 | 0 | 0 | 0 | 0 | 0 | 0 | 0 | 0 | 0 | 0 | 0 | 0 | 0 | 0 | 0 | 0 | 0 |
| 2 | 10 | 0 | 0 | 0 | 0 | 0 | 0 | 0 | 0 | 0 | 0 | 0 | 0 | 0 | 0 | 0 | 0 | 0 |
| 3 | 0 | 22 | 0 | 0 | 0 | 0 | 0 | 0 | 0 | 0 | 0 | 0 | 0 | 0 | 0 | 0 | 0 | 0 |
| 4 | 0 | 14 | 0 | 0 | 0 | 0 | 0 | 0 | 0 | 0 | 0 | 0 | 0 | 0 | 0 | 0 | 0 | 0 |
| 5 | 0 | 0 | 5 | 0 | 0 | 0 | 0 | 0 | 0 | 0 | 0 | 0 | 0 | 0 | 0 | 0 | 0 | 0 |
| 6 | 0 | 0 | 0 | 10 | 0 | 0 | 1 | 0 | 0 | 0 | 0 | 0 | 0 | 0 | 0 | 0 | 0 | 0 |
| 7 | 0 | 0 | 0 | 0 | 9 | 0 | 0 | 0 | 0 | 0 | 0 | 0 | 0 | 0 | 0 | 0 | 0 | 0 |
| 8 | 0 | 0 | 0 | 0 | 0 | 10 | 0 | 0 | 0 | 0 | 0 | 0 | 0 | 0 | 0 | 0 | 0 | 1 |
| 9 | 0 | 0 | 0 | 0 | 0 | 0 | 4 | 0 | 0 | 8 | 0 | 0 | 0 | 0 | 0 | 0 | 0 | 0 |
| 10 | 0 | 0 | 0 | 0 | 0 | 0 | 0 | 0 | 0 | 0 | 0 | 0 | 0 | 0 | 0 | 0 | 2 | 0 |
| 11 | 0 | 0 | 0 | 0 | 0 | 0 | 0 | 1 | 7 | 0 | 0 | 0 | 0 | 0 | 0 | 0 | 0 | 0 |
| 12 | 0 | 0 | 0 | 0 | 0 | 0 | 0 | 0 | 0 | 0 | 6 | 0 | 0 | 0 | 0 | 0 | 0 | 0 |
| 13 | 0 | 0 | 0 | 0 | 0 | 0 | 0 | 0 | 0 | 0 | 0 | 16 | 0 | 0 | 0 | 0 | 0 | 0 |
| 14 | 0 | 0 | 0 | 0 | 0 | 0 | 0 | 0 | 0 | 0 | 0 | 0 | 6 | 0 | 0 | 0 | 0 | 0 |
| 15 | 0 | 0 | 0 | 0 | 0 | 0 | 0 | 0 | 0 | 0 | 0 | 0 | 0 | 7 | 0 | 0 | 0 | 0 |
| 16 | 0 | 0 | 0 | 0 | 0 | 0 | 0 | 0 | 0 | 0 | 0 | 0 | 0 | 0 | 11 | 0 | 0 | 0 |
| 17 | 0 | 0 | 0 | 0 | 0 | 0 | 0 | 0 | 0 | 0 | 0 | 0 | 0 | 0 | 0 | 3 | 2 | 0 |

minSNPs algorithm. The minSNPs' requirement in percent mode that SNP sets provide 100% sensitivity for the group of interest is very stringent. A false negative defined by a single member of a group of interest disqualifies a position from inclusion in a SNP set. Being able to capture more diversity for the analysis by using the transformation procedure appears to have been critical, with this being explicable because positions with ambiguity codes will likely be the most diverse. A possible workaround for this constraint on SNP selection is to run separate analyses, each with subsets of the group of interest. This could yield SNPs that provide low but non-zero false negatives with respect to the entire data set.

## Derivation of SNP sets with "Simpson" mode

We then used minSNPs to derive the Simpson's index of diversity-maximised SNP sets from the *P. vivax* alignment. Both the approaches described above for accommodating ambiguity codes were used. Five SNP sets, each comprising five SNPs, were derived using each approach. When all the positions that encompassed at least one ambiguity code were excluded from the analysis, the index of diversity values obtained were 0.751, 0.750, 0.572, and 0.564 (two sets). The most discriminatory SNP set (index of diversity = 0.751) was investigated further. It was determined that the matrix defined eight allelic profiles. Although this number of profiles and the index of diversity do not indicate high discrimination, there was close concordance between allelic profile and country of origin (Table 3). Thus, within the context of the diversity defined by the input matrix, five SNPs can accurately reveal *P. vivax* country of origin. Complete results in this figshare: https://doi.org/10.6084/m9.figshare.19580299.v1. Similar results were obtained with 80% of the sequences, chosen at random, shown in this figshare: https://doi.org/10.6084/m9.figshare.19580299.v1.

When the analysis was repeated with the transformed ambiguity codes, very different results were obtained. The index of diversity values were from 0.958 to 0.960, which is considerably higher than in the previous experiment. Consistent with this, the SNP sets defined 31-32 allelic profiles. The numbers of specimens defined as untypeable were significant, ranging from 64 to 68 (25%–26% of specimens). The concordances with country of origin were poor. Even with the larger number of allelic profiles, there were numerous instances of specimens from different countries having the same profile. A likely explanation is that positions that encompass ambiguity codes are polymorphic within countries. Such SNPs are more likely to generate ambiguity codes because both alleles may be present in a mixed infection. The exclusion of these positions will enrich for SNPs that separate specimens from different countries and are monomorphic within countries. This would be expected to facilitate the derivation of SNP sets that indicate the country of origin. The complete results are in this figshare: https://doi.org/10.6084/m9.figshare.19580299.v1. Similar results were obtained with 80% of the sequences, chosen at random, when in this figshare: https://doi.org/10.6084/m9.figshare.19580299.v1. Scripts written in the course of this arm of the study are shown in this figshare: https://doi.org/10.6084/m9.figshare.19580299.v1. A graphical representation of the results is in this figshare: https://doi.org/10.6084/m9.figshare.19580299.v1.

**Table 3** ***P. vivax* genotypes defined by high-diversity index SNP set 1 (ambiguity codes excluded *vs.* substituted).** (A) The SNP set was derived from a matrix where all positions that encompassed an ambiguity code were excluded from the analysis. The index of diversity is 0751. The SNP positions are 340505 (Chromosome 13), 460741 (Chromosome 12), 854772 (Chromosome 10), 531315 (Chromosome 6), 2100572 (Chromosome 12). The SNP numbering represent the relative position of the SNPs within the chromosome. (B) The SNP set was derived from a matrix where the ambiguity codes were transformed into the major allele at that position. The index of diversity is 0.960. The SNP positions are 1269895 (Chromosome 14), 1240935 (Chromosome 13), 1812716 (Chromosome 11), 1717060 (Chromosome 9), 1141805 (Chromosome 10).

| Genotype | Malaysia | Thailand | Indonesia | Imported |
|---|---|---|---|---|
| | | A. Ambiguity codes excluded | | |
| 1 | 26 | 0 | 0 | 0 |
| 2 | 17 | 1 | 0 | 0 |
| 3 | 3 | 3 | 0 | 0 |
| 4 | 1 | 91 | 0 | 1 |
| 5 | 0 | 9 | 0 | 0 |
| 6 | 1 | 0 | 80 | 2 |
| 7 | 0 | 0 | 11 | 0 |
| 8 | 0 | 0 | 9 | 0 |
| 9 | 0 | 0 | 3 | 0 |
| 10 | 0 | 0 | 1 | 0 |
| | | B. Ambiguity codes transformed | | |
| 1 | 26 | 0 | 0 | 0 |
| 2 | 5 | 0 | 1 | 0 |
| 3 | 3 | 5 | 0 | 0 |
| 4 | 3 | 0 | 2 | 1 |
| 5 | 2 | 0 | 5 | 0 |
| 6 | 1 | 7 | 0 | 0 |
| 7 | 1 | 5 | 0 | 0 |
| 8 | 1 | 0 | 4 | 0 |
| 9 | 0 | 8 | 0 | 0 |
| 10 | 0 | 8 | 0 | 0 |
| 11 | 0 | 7 | 0 | 0 |
| 12 | 0 | 6 | 0 | 0 |
| 13 | 0 | 5 | 0 | 1 |
| 14 | 0 | 5 | 0 | 0 |
| 15 | 0 | 5 | 0 | 0 |
| 16 | 0 | 5 | 0 | 0 |
| 17 | 0 | 3 | 0 | 0 |
| 18 | 0 | 3 | 0 | 0 |
| 19 | 0 | 2 | 0 | 0 |
| 20 | 0 | 1 | 6 | 0 |
| 21 | 0 | 1 | 4 | 0 |
| 22 | 0 | 1 | 0 | 0 |
| 23 | 0 | 0 | 6 | 0 |

**Table 3** (*continued*)

| Genotype | Malaysia | Thailand | Indonesia | Imported |
|----------|----------|----------|-----------|----------|
| 24 | 0 | 0 | 6 | 0 |
| 25 | 0 | 0 | 6 | 0 |
| 26 | 0 | 0 | 5 | 0 |
| 27 | 0 | 0 | 5 | 0 |
| 28 | 0 | 0 | 5 | 0 |
| 29 | 0 | 0 | 4 | 0 |
| 30 | 0 | 0 | 4 | 0 |
| 31 | 0 | 0 | 4 | 0 |
| 32 | 0 | 0 | 3 | 1 |

Thus for *P. vivax*, diversity-maximised SNPs without ambiguity codes are useful as minimal sets of markers for geographical tagging (*Adam et al., 2022*). Conversely, including ambiguity codes yields SNP sets better for rapid screening for epidemiological linkage on small scales of time/space.

## Derivation of SNP sets from multiple BioProjects

We further demonstrated the ability of minSNPs to analyse large datasets. The detailed procedure and complete results for these experiments are in figshare: https://doi.org/10.6084/m9.figshare.19582885.v1. Within the supplementary information, the procedure and results are presented concisely in the supplementary overview. Large data sets are in the other files.

We obtained *S. aureus* short read data in fastq format from NCBI BioProjects PRJEB3174 (*Toleman et al., 2016*; *Coll et al., 2020*), PRJEB32286 (*Coll et al., 2020*), and PRJNA400143 (*Manara et al., 2018*)). These data and the STARRS fastq data (BioProject: PRJEB40888) were used to create an orthologous SNP matrix using a modified SPANDx pipeline (*Sarovich & Price, 2014*). The modified pipeline and the detailed procedures are shown in figshare: https://doi.org/10.6084/m9.figshare.19582885.v1 in the Overview file.

The matrix encompasses 3,279 isolates (including the reference genome Mu50) and 164,335 SNP positions, and is in figshare: https://doi.org/10.6084/m9.figshare.19582885.v1. The mapping to the reference genome is in this figshare: https://doi.org/10.6084/m9.figshare.19582885.v1. We used this to validate the SNPs discriminating both ST762 and CC1 obtained earlier using only the STARRS dataset. Apart from one SNP set, all the previously identified single SNP sets retained 100% sensitivity and specificity for ST762 with this large data set. However, two of the SNPs were not present in the matrix. For CC1 (ST1, ST762, ST2851, ST2981), most of the previously identified SNP sets were not fully present in the matrix (*i.e.,* the STARRS derived sets often included positions that were not included in the merged matrix due to quality filtering). For similar reasons, not all the members of the previously identified high Simpson's index SNPs-sets were present in the new matrix, and no meaningful comparison between the previous analysis and current analysis could be made. The results are in this figshare: https://doi.org/10.6084/m9.figshare.19582885.v1.

### Derivation of SNP sets to discriminate ST762 and CC1 with "percent" mode

We re-ran the same tasks in previous STARRS datasets with the matrix. We identified 50 individual SNPs and 50 two-member SNP sets that discriminate all ST762 isolates from all others. The results are in this figshare: https://doi.org/10.6084/m9.figshare.19582885.v1. We similarly identified 39 individual SNPs and 61 two-member SNP sets (100 SNPs sets) that discriminate all CC1 isolates from all others. The results are in this figshare: https://doi.org/10.6084/m9.figshare.19582885.v1.

### Derivation of SNP sets with "Simpson" mode

We then experimented with the Simpson mode analysis to accomplish two different tasks. First, we attempted to identify SNPs that discriminated all CCs from each other. To accomplish this, all the variant positions between isolates within the same CC were identified and recorded. A reduced matrix was then constructed that contained only a single isolate from each of the CCs. We then excluded from analysis all the previously recorded variant positions within CCs before running a Simpson mode search. It was found that a minimum of seven SNPs were required to discriminate all 33 CCs from each other. minSNPs was tasked to provide 200 alternative SNP sets that achieved a diversity value of 1.0. Of these, 165 of the sets had seven members; the remaining had eight members. The results are in figshare: https://doi.org/10.6084/m9.figshare.19582885.v1.

Next, we explored the resolving power of SNP sets identified simply to maximise $D$ without reference to CC. Similarly, we identified five 10-SNP sets with a high Simpson's index of diversity (figshare: https://doi.org/10.6084/m9.figshare.19582885.v1 Overview). Prior to running the minSNPs analysis, all but a subset of 100 CC22 isolates were randomly selected to be included in the input matrix to avoid overly biasing the analysis to include SNPs that discriminated within CC22. We obtained SNP sets with diversity values (recalculated using the entire matrix) ranging from 0.6314 to 0.6461. We selected the SNP set with the highest diversity value and constructed the allelic profile with the first five SNPs. As expected from the similar experiment performed with the smaller STARRS data set, there was close but imperfect correspondence between CC and allelic profile, even though there was no reference to CC in the SNP derivation procedure. The results are included in this figshare: https://doi.org/10.6084/m9.figshare.19582885.v1 Overview, from page 9. A graphical representation of the results is in this figshare: https://doi.org/10.6084/m9.figshare.19582885.v1.

## CONCLUSIONS

minSNPs is a new R package. This software provides a flexible means for deriving from comparative genome data SNP sets that are optimised for lineage-specific or generalised resolving power. The functionality of minSNPs has been demonstrated using large genome-wide orthologous SNP matrices from *S. aureus* and *P. vivax*. minSNPs can facilitate the derivation of genetic marker sets customised for defined surveillance applications from global-scale genomic diversity data.

## ACKNOWLEDGEMENTS

The authors thank Mariana Barnes and Tegan Harris from the Menzies School of Health Research for assistance with the installation of minSNPs onto the Charles Darwin University high performance computer (HPC) cluster and also with the associated software documentation tasks. The authors thank Kamil Braima, Angela Rumaseb and Aiden Webb for testing the documentation.

### Funding

Kian Soon Hoon (as student) and Philip Giffard, Deborah Holt and Sarah Auburn (as the supervisory team) are recipients of a Charles Darwin University "Charles Darwin International PhD Scholarship" to pursue this project. The early stages of this project were supported by a Charles Darwin University Institute of Advanced Studies Rainmaker Startup Grant, (ID 18916864), awarded to Philip Giffard and Peter Shaw. The funders had no role in study design, data collection and analysis, decision to publish, or preparation of the manuscript.

### Grant Disclosures

The following grant information was disclosed by the authors:
Charles Darwin University "Charles Darwin International PhD Scholarship".
Charles Darwin University Institute of Advanced Studies Rainmaker Startup Grant: 18916864.

### Competing Interests

The authors declare there are no competing interests. Peter Shaw is employed by Oujian Laboratory.

### Author Contributions

- Kian Soon Hoon conceived and designed the experiments, performed the experiments, analyzed the data, prepared figures and/or tables, authored or reviewed drafts of the article, and approved the final draft.
- Deborah C. Holt conceived and designed the experiments, analyzed the data, authored or reviewed drafts of the article, and approved the final draft.
- Sarah Auburn conceived and designed the experiments, authored or reviewed drafts of the article, and approved the final draft.
- Peter Shaw conceived and designed the experiments, authored or reviewed drafts of the article, and approved the final draft.
- Philip M. Giffard conceived and designed the experiments, analyzed the data, authored or reviewed drafts of the article, and approved the final draft.

### Data Availability

The MinSNPs software and documentation is available in Cran, Github and Zenodo:

- https://github.com/ludwigHoon/minSNPs

- https://cran.r-project.org/package=minSNPs

- Hoon, Kian Soon. (2023). MinSNPs (0.0.2). Zenodo. https://doi.org/10.5281/zenodo.7618983.

The external data used are available at:

- STARRS matrix from Holt DC, Harris TM, Hughes JT, Lilliebridge R, Croker D, et al. (2021) Longitudinal whole-genome based comparison of carriage and infection associated *Staphylococcus aureus* in northern Australian dialysis clinics. PLOS ONE 16(2): e0245790. https://doi.org/10.1371/journal.pone.0245790

- *P. vivax* data, Malariagen: https://www.malariagen.net/resource/24

- Bioprojects used to construct a *S. aureus* mega alignment: PRJEB3174, PRJEB32286, PRJNA400143, PRJEB40888

Additional information is available at figshare:

- Hoon, Ludwig (2023): minSNPs_runtime(supp.1). figshare. Dataset. https://doi.org/10.6084/m9.figshare.19579816.v1

- Hoon, Ludwig (2023): STARRS_analysis(supp.2). figshare. Dataset. https://doi.org/10.6084/m9.figshare.19579837.v1

- Hoon, Ludwig (2023): Supplementary_Methods_Blastn(supp.3).docx. figshare. Online resource. https://doi.org/10.6084/m9.figshare.19579831.v1

- Hoon, Ludwig (2023): Supplementary_method_result_vivax(supp.4). figshare. Online resource. https://doi.org/10.6084/m9.figshare.19580299.v1

- Hoon, Ludwig (2023): Megaalignment(Supp.5). figshare. Online resource. https://doi.org/10.6084/m9.figshare.19582885.v1

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
