# Peer review of "minSNPs: an R package for the derivation of resolution-optimised SNP sets from microbial genomic data"

_PeerJ, doi:10.7717/peerj.15339_

## Round 0.1 · original submission · Major Revisions

Please see the reviewers' comments below, in particular those from Reviewer 2.

Reviewer 1 ·

Basic reporting

In this work authors present minSNPs, a R package to derive sets of polymorphisms from sequence alignments based on high combinatorial discriminatory power. I found the result presented are interesting and overall the manuscript is well structured. However, more details regarding the package, its usage and discussion regarding potential limitations could complement this manuscript to better known when use this new R package in a specific dataset.

Experimental design

Authors mentioned that this package is a re-develompent of Minimun SNPs, which improve the flexibility, error handling and output formats. However, the comparisons and discussion of these features are not clearly proposed. Moreover I would suggest to compare and clarify the advantages of this package with conventional tools or packages developed for SNPs identification.

Authors mentioned they developed a pipeline to generate minSNPs input files from multiple short reads data sets, however it is not clear how this pipeline works. I suggest including a figure to clarify and help to understand how to prepare the input data.

Please give more information regarding the processor and memory of the laptop used, this will give more information for the future users of this package. Moreover, authors could explicitly say what are the number of cores and memory needed for a good performance of the package.

Figure S3 is not available at the link, please review it.

Validity of the findings

Author mentioned they found some inconsistencies when the analysis was repeated with the transformed ambiguity codes. How could authors ensure this is a problem of the dataset and not a problem of the package design? Do you think this package could work better for a specific type of SNPs profiles? if it is, please discuss which ones would be the appropiate datasets to use this package.

Reviewer 2 ·

Basic reporting

The manuscript is well written grammatically and it is clear that a lot of effort has gone into the work. However, the program and the analyses are presented in a very nebulous way that is difficult to understand. However, improved organization could provide greater clarity and additional information could be added to introduction. Figures and Tables could be updated for clarity. It would truly be helpful to more clearly and concisely present the algorithm and findings. Additional headings to organize the results further could be of aid. Frankly, data presented in figures was not enough to assess and then supplemental data was a barrage of multiple figures and tables asking the reader to assess all the data for themselves at a granular level. Better guiding the reader concretely would be much appreciated.

Experimental design

The authors present an R package that is designed to take an input set of SNPs from bacterial or small protozoal genomes and detect a small subset of SNPs that defines a subgroup/lineage of interest or to general discriminant the individuals within a population. The program is an implementation of the key algorithms from an older Java program Minimum SNPs which has been ported to R programming language and further improved. They provide several use cases looking at bacteria and protozoal genomes. The protzoal application is particularly interesting given it is a eukaryotic organism that recombines and presumably can rapidly shuffle its genetic markers unlike bacteria. t would truly be helpful to more clearly and concisely present the algorithm and findings highlighting how the algorithm is different than Minimum SNPs. Diagrams outline the work flow and algorithm I think would aid the reader to get a concise snap shot.

Abstract does not mention that this is a rewrite/port of Minimum SNPs – already existing software. Clarifying in the abstract as well as in the main text more precisely the exact differences would be of great utility for readers. For instance, Lines 88-90 : “improvement of flexibility, error handling, and output formats” is not clearly defined in introduction. No specifics in truly why the need for the new port or what flexibility, error handling, output format improvements are? Was there a dataset or analysis that could not be accomplished with Minimum SNPs? How is it easier to use? There is not adquate compare and contrast to help a reader choose this program over the other.

Table 1 is inadequate in terms of exploring runtime determinants. The reader should not have to try to dig through a complicated supplemental file to see if runtimes are linear with all the stated parameters. Presumably, # of sequences and # of SNP is the major determinant which does not even vary in Table 1.

In general, the supplemental data is overwhelming. Simpler summaries of the supplemental data should be used to make the points in the text. Having a supplemental link go to tens of files makes the readers job difficult as well as the reviewer’s difficult in terms of evaluation. Even the main tables are complicated. Table 3: has multiple tables within which is non standard?

Unclear what the utility is of working with ambiguous codes since they are ambiguous and not good markers presumably? The ambiguous codes are all presumably due to mixed infections with two alleles . Unless parasites are predominantly clonally transmitted recombination will destroy the utility of a minimum set of SNPs once recombination occurs.

It is unclear why a user can’t simply import a vcf or another standard file format. It seems malaria data was turned from a vcf into a fasta format with ambagious calls? Why not have a simple function to import a vcf.

A discussion of the effects of recombination? Are the SNP set all on the same chromosome or different chromosomes for malaria? What will subsequent recombination due to the utility if they are on different markers?

Validity of the findings

Validity of the findings appears sound although issues above do not allow full assessment. Underlying data not fully explored given not well presented. Conclusions could be broader with more discussion. It would also be useful to better separate the results from the discussion or label the results – results and discussion. Frankly the conclusion felt inadequate summary

Additional comments

Queries

Line 48: what type of alignment? Pairwise or presumably multiple alignment? How big a set of SNPs? Does the alignment have to contain the full sequence? Or just variant regions? How as a larger genome like malaria dealt with? The options should be fully discussed here vs later.

Line 50: multiple: bio-projects? What are these? Two different multiple alignments that yo internally merge?

Line 52: what is input volume – length of alignments ? #of alignments? Probably should assess as individual parameters???

Line 76: What is the scale of the # of SNPs? 100s-1000s more or less?

Line 80: what were the issues with Minimum SNPs requiring a rewrite? No specifics offered.
Line 88-90: again. it is unclear – very vague the reasons for the redevelopment?
Line 98: It would be nice to have an overview of the method as opposed to starting with the format of the input? Perhaps a diagram would be a useful overview to properly oreint the reader (and contrast to Minimum SNPs as a supplemental?
Line 100 FASTA alignment ? Presume multiple vs pairwise? Why can’t a user input a genotyping matrix or vcf? Fasta alignment seems unwieldy beyond bacteria?
Line 104: how many SNPs are outputted?
Line 127: This paragraph does not clearly compare and contrast “minimum SNPs” from minSNPs
Line 149-154: clearly state if these comparisons are part of the output?
Line 155: These functions unclear as to what these functions are?
Line 169: Shouldn’t that be presented in the methods? A pipeline for orthologous SnP matrices
Line 186: Where does the matrix come from? Also use () for numbering not punctuation . that can be interpreted as a full stop in a sentence
Line 220: This is interesting using simpson but curious what would happen varying the # of SNPs or is this fixed???? Does simpson index improve with additional snps? Could varying this be informative for the reader.
Line 233: How was this data entered was it fasta multiple alignment that is required given you are likey starting with a vcf??
Line 256: genomic clustering patterns not defined?
Line 262: How do you determine the major allele? Is this the population major allele which would be fortuitous (Line 264) or the strain major allele which would be deterministic?
Line 266-271: It is unclear if this is what is applied or prior information in “this approach”). I would suggest subheadings for the different approaches to vivax (and above staph analyses that clearly state how ambiguous codes were dealt with.
Line 305-308 : unclear that there are fixed differences between populations….
Line 322: obviously there is another way to input data apart from fasta alignments.

---

## Round 0.2 · Minor Revisions

I am happy with the science in this paper, however I think we need to check the presentation with regards to the data submitted.

The inclusion of lots of data as supplementary data needs to be made a little easier. At the moment some supplementary are referenced in the text by a Figshare URLs whereas others are referenced with text like "The complete results are in this file", this is not consistent. Also I would imagine that an interested reader would want to download all the data files and then work through the paper that way however it would not be easy to link "This file" to the actual specific table without the clickable URL. I would recommend relabelling all examples of "this file" to a specific name e.g. Supplementary Fig 1, this also seems more like standard practice. Also I don't see any legends to the Supplementary data, I think much of it would benefit from some descriptive text to enable the reader to better understand.

Also finally to add the journal require any externally linked data to have a DOI on publication, so these should probably also be included in the paper rather than Figshare URL's. I would guess the journal won't want URLs in the text either.

Some specific suggested edits based on the word version line numbers:

Line 50: minSNPs is a proper noun so should start the sentence with a lower case "m" ie minSNPs
Line 189: Staphylococcus aureus already written out in full, so should be S. aureus
Line 190: same for P. vivax
Line 205: Replace the "PCs at Charles Darwin University" with the important spec i.e. cpus and RAM. we have no way of knowing how good those PCs are
Line 280: need to explain "High-D" somewhere

---

## Round 0.3 · accepted · Accept

Thank you for addressing the concerns of the reviewers.